# Is Childhood Overweight/Obesity Perceived as a Health Problem by Mothers of Preschool Aged Children in Bangladesh? A Community Level Cross-Sectional Study

**DOI:** 10.3390/ijerph16020202

**Published:** 2019-01-12

**Authors:** Mohammad Sorowar Hossain, Mahbubul H. Siddiqee, Shameema Ferdous, Marzia Faruki, Rifat Jahan, Shah Md. Shahik, Enayetur Raheem, Anthony D. Okely

**Affiliations:** 1Biomedical Research Foundation, Dhaka 1216, Bangladesh; mahbubul.siddiqee@brfbd.org (M.H.S.); shameema.ferdous@brfbd.org (S.F.); farukiphysio@gmail.com (M.F.); rajahan@gmail.com (R.J.); sm.shahik@brfbd.org (S.M.S.); enayetur.raheem@brfbd.org (E.R.); 2Department of Environmental Management, Independent University, Dhaka 1229, Bangladesh; 3Department of Mathematics and Natural Sciences, BRAC University, Dhaka 1212, Bangladesh; 4Department of Public Health Sciences, University of North Carolina at Charlotte, Charlotte, NC 28223, USA; 5Early Start, Faculty of Social Sciences, University of Wollongong, Wollongong, NSW 2500, Australia; tokely@uow.edu.au

**Keywords:** childhood, overweight, obesity, maternal perception, Bangladesh

## Abstract

Childhood obesity is rapidly rising in many developing countries such as Bangladesh; however, the factors responsible for this increase are not well understood. Being the primary caregivers of children, particularly in developing countries, maternal perceptions and knowledge could be important factors influencing the weight status of children. This study aimed to assess maternal perceptions of childhood obesity and associated socio-demographic factors in Bangladesh. A cross-sectional study using stratified random sampling was conducted among 585 mothers whose children aged 4 to 7 years attended preschools in a district town. Body Mass Index of the children was calculated and weight status categorized according to the Centers for Disease Control (CDC) criteria. Maternal perceptions were assessed using a self- or interviewer-administered questionnaire. Multinomial logistic regression was used to obtain crude and adjusted odds ratios. Fourteen percent of children were overweight or obese and approximately 30% were underweight. Only 3.1% of children were perceived as overweight/obese by their mothers. Over one-third (35%) of mothers perceived that childhood overweight/obesity could be a health problem and over two-thirds (68.6%) were not aware of any health consequences of childhood obesity. Maternal perceptions were significantly associated with maternal education, family income, and weight status of the child but were not associated with the sex of the child. We have identified knowledge gaps regarding maternal perception of childhood obesity and its contributing factors in a developing country. These findings can be used to develop and test parent-focused educational interventions for preventing childhood obesity in Bangladesh.

## 1. Introduction

Many developing countries with high rates of undernutrition have witnessed a rapid rise in the prevalence of childhood overweight and obesity over the past 20 years [1]. Obesity among children has emerged as a serious public health challenge [2]. A tenfold increase (11 million to 124 million) in child and adolescent obesity worldwide has been documented over a period of 40 years (1976–2016), [1].

Bangladesh, a lower middle-income country with a population of over 160 million, has undergone demographic, nutritional and epidemiological transitions over last two decades [3]. In 1989–1990, the prevalence of underweight was 65% among children of under five years of age [4]. Despite significant progress since then, the prevalence of undernutrition remains high, at around 33% according to the 2014 Bangladesh Health and Demographic Survey [5]. Concurrently, childhood overweight and obesity has increased from 3.6% in 1998–2003 to 5.7% during 2004–2009 to 7.9% in 2010–2015 [6]. While childhood obesity is an emerging health problem in Bangladesh, the factors responsible for this increase are poorly understood with limited research in this area.

This increase in the prevalence of childhood obesity in Bangladesh may be attributed to changes in urbanisation and low levels of parental education about the risk factors for childhood obesity. Bangladesh is undergoing rapid urbanisation and the majority of the population will live in urban areas by 2040 [3]. Bangladesh has been successful in reducing the rate of child undernutrition over the last two decades [7]. While reduction in undernutrition for children has been achieved, the concurrent rise in childhood obesity has introduced a double-burden of under and overnutrition in the same country.

Mothers have major roles in shaping the knowledge, behaviours, and attitudes, in their children during the early ages [8,9]. Mothers’ roles have become more important when they are the primary caregivers; especially in developing countries like Bangladesh. Therefore, maternal perceptions about their child’s health have an important influence on children’s nutrition and physical activity. Perception is a multifaceted expression which is highly influenced (if not determined) by knowledge and cultural practices and beliefs of any individual [10,11]. Understanding maternal knowledge and perceptions is an important step in identifying potential intervention strategies for promoting healthy weight among children.

To our knowledge, no study has examined parental perceptions towards children’s weight status in Bangladesh. The aims of this study were to investigate (i) maternal perceptions of childhood obesity and its contributing factors (ii) the difference between actual and perceived weight status and (iii) what socio-demographic factors are associated with maternal perceptions of childhood obesity. We hypothesized that more than 20% of the mothers would consider childhood obesity as a health problem and this maternal perception would differ by school type. Our secondary hypothesis was that, there would be difference between perceived and actual overweight/obesity status of children by their mothers.

## 2. Materials and Methods

### 2.1. Study Design and Setting

This was a cross-sectional study conducted in the Jamalpur district town (population: 142,764, area: 53.45 km^2^) of Bangladesh [12]. Jamalpur is one of 64 administrative districts in Bangladesh and located 140 km northwest of the capital city, Dhaka. Typically, each district is comprised of an urban area (named as a district town or municipality), which is the centre of all administrative activities of the district. Apart from few major cities all district towns are relatively homogeneous in the context of language, food habit, cultural norms and level of urbanization (Appendix A).

### 2.2. Participants

The target population was all the children aged four to seven years attending preschools in the municipality of Jamalpur. In Bangladesh, this includes playgroups, nurseries, kindergarten-I (KG-I) and KG-II. The study population included secular preschools, which are usually a part of primary school and *Noorani Madrasas*- religious preschools equivalent to preschools. Unlike secular counterparts, most Noorani Madrasas are not a part of primary school. Families with relatively higher socio-economic backgrounds prefer secular preschools for their children, while highly conservative Muslim families prefer Noorani Madrasa preschools. Tuition fees in religious schools are nominal (often free of cost), whereas studying in the secular schools incur varying levels of tuition fees. Considering the secular and religious preschools as two strata in the population, a stratified random sample was employed. A replacement preschool was selected randomly from the remaining preschools if a selected preschool declined to participate. The final unit of analysis was students and their mothers in the selected preschools/Madrasas. If a mother had two or more children studying at the same preschool, measurements for the youngest child were taken. Ethical approval for this study was obtained from the Ethical Review Board of Biomedical Research Foundation, Bangladesh (Memo no: BRF/ERB/2018/003). Informed written consent was taken from participating mothers before data collection.

### 2.3. Sample Size

To obtain an estimate of the proportion of mothers indicating that childhood obesity was a problem within 5% margin of error with 99% confidence, a sample size of 470 was required. The sample size was calculated using R package sample sizes 4 surveys [13]. For this, a two-stage complex design with 53 clusters (schools) in the population with approximately 1600 students, 20% of mothers being aware of childhood obesity as a problem and an intraclass correlation coefficient of 0.02 (i.e., design effect of 1.6) were assumed. To obtain the minimum sample size required, eight secular schools and four Noorani Madrasas were selected according to probability proportional to the stratum size.

### 2.4. Variables and Procedures

Questionnaire: Maternal perceptions were assessed using a structured questionnaire that was drafted based on existing literature and expert opinion. The questions formed part of a larger survey examining demographic patterns, maternal perceptions, and the potential determinants of childhood obesity. In this paper, only data on demography and maternal perception were reported. Survey questions asked mothers for their age, education, occupation, marital status, husband’s education and occupation and monthly family income. Mothers were asked if childhood obesity was an health issue (response categories: ‘a serious problem’, ‘an ordinary problem’, ‘not sure if it is a problem’, ‘not a serious problem’, ‘not a problem at all’). Other perception-related questions (response categories: yes, no, do not know) included: (i)whether they considered that overweight/obesity during childhood was a sign of good health; (ii) whether they thought children who were overweight/obese during childhood might have good health when they grow up; (iii) whether they perceived the following factors as reasons behind increases in childhood obesity: consuming junk food/fast food and soft drinks, decreased participation in physical activity and spending more than two hours watching TV/using smartphones/playing video games. Mothers were also asked what might be some of the consequences of childhood obesity (response options: diabetes, heart disease, psychological problems, do not know, others) and whether they felt stressed that their relatives might blame them for their child being lean (response categories: yes, no, have not thought about it).

Data collection: The survey was conducted within the preschool premises in presence of teachers nominated by the respective preschools. Data were collected by volunteers specially trained for this study. On the day of survey, all the participating mothers were requested to gather at the preschool. Mothers were provided a detailed explanation on how to complete the survey. In cases (~12% of all participants) where the mothers had difficulty in reading the instructions and completing the questionnaire, they were separated and one-on-one assistance was provided by a member of the study team. After completing the questionnaire, the mothers brought their children to the makeshift measuring booths for height and weight measurements. A total of 36 booths were set up. Each of the booths was attended by at least two members from the survey team; one trained in taking height and weight while the other member recorded the measurements. All the measuring instruments were checked on each sampling day (prior to taking measurements) and calibrated as necessary.

Measurement of height and weight: Measurement of height was taken using a stadiometer (Seca, Hamburg, Germany) with unit of 0.1 cm. Weight of each participant was measured using digital weighing scales with unit of 0.01 kg (Mega, Dhaka, Bangladesh). Height and weight measurements were used to calculate body mass index (BMI) of the child using the Centre Disease Control (CDC, Atlanta, GA, USA) standards [14].

### 2.5. Statistical Analyses

Descriptive and inferential statistical procedures were used to analyze the data. Sample statistics such as means and proportions (percentages) were calculated. For testing association between categorical variables, Pearson’s chi-square test with continuity correction was used. For variables with small cell counts, Fisher’s exact test was used. Multinomial logistic regression was used to obtain crude and adjusted odds ratios.

The BMI of a child was calculated by dividing the weight in kilograms by the height in metres squared [14]. BMI percentiles were calculated using CDC’s LMS (Lambda-mu-sigma/Learning management System) parameters as reference [15], allowing comparison of BMI by different ages and gender [16]. The LMS parameters were the median (M), the generalized coefficient of variation (S), and the power in the Box-Cox transformation (L) [17]. The R package “childsds” was used to obtain the BMI percentiles for the given age and sex of the child [18].

CDC Guidelines were used to classify each child into four different weight categories. For children two years or older, the BMI < 5th percentile was considered underweight; between 5th percentile and ≤ 85th percentile was considered healthy/normal weight; between 85th and <95th percentile overweight; and ≥95th percentile was considered obese.

## 3. Results

### 3.1. Respondent Characteristics

A total of 649 mothers participated in the survey. After screening completed questionnaires, 64 (9.9%) cases were removed from the analysis because of incomplete data resulting in 585 mothers in the final sample. Half the mothers (51.7%) had at least 12 years of education, while 64.1% of the fathers had a similar level of education. Almost all the mothers were housewives (87.1%) and lived with their husband (96.6%). Nearly 66% of the households comprised up to two children and around 60% of all households had a monthly income of less than BDT 25,000 (USD $312) (Table 1). Mothers of secular background had significantly higher levels of education and family income compared with their religious (Noorani) counterparts. (Table 1).

### 3.2. Prevalence of Childhood Overweight/Obesity

Nearly one third (30.4%) of the children were underweight while 14% of the children were overweight or obese (Table 2). The prevalence of obesity was nearly twice as high in boys than in girls (10.26% vs. 5.66% respectively). The proportions of overweight/obese children in Noorani and secular preschools were 10.7% and 15.7%, respectively. The proportion of healthy weight children was 59% and 54% in Noorani and secular preschools, respectively. The proportion of obese children was almost twice as high in secular preschools than in Noorani preschools (9.9% vs. 4.8%).

### 3.3. Maternal Perception of Childhood Overweight/Obesity

Table 2 reports the mothers’ perceptions of their child’s weight status. About 3% of children were perceived as overweight/obese by their mothers, a figure that was much lower than the actual measured prevalence of overweight/obesity (14%, Fisher’s exact test, *p* < 0.001).

Compared with the actual weight status, mothers were able to recognize their underweight children correctly.

Table 3 reports maternal perception of childhood obesity and knowledge of factors contributing to childhood obesity. Just over one third (35.04%) of respondents perceived that childhood overweight/obesity could be a health problem for their children, which was significantly higher than our hypothesis of 20% (Chi-squared = 84.82, *p* < 0.001). This perception did not differ between two school types (Chi-square = 0.38, *p* = 0.5) (Appendix A). More than two third of mothers did not know any consequences of childhood obesity (68.6%). Similarly, a high proportion of mothers could not see the link between childhood obesity and risk factors such as consuming unhealthy food (77.5%), low levels of physical activity (62.4%), high levels of screen time (71.3%) and lack of outdoor playing spaces (56.2%). There were no statistically significant differences in perceptions and knowledge (except screen time and lack of playground) regarding overweight/obesity among mothers from secular or Noorani backgrounds Appendix A).

The differences in maternal perceptions of childhood obesity and the contributing factors between parents of boys and girls, low and high education and income levels, and of children of different weight statuses are reported in Appendix A. These differences in proportions were tested and the unadjusted and adjusted odd ratios (AOR) are reported in Table 4 and Table 5. Compared with mothers from low-income families, mothers from high-income families were less likely to think that an obese child would be healthy as an adult (AOR: 0.42, 95% CI: 0.18–0.94). Compared with parents whose child had a healthy weight, parents whose child was underweight (AOR: 0.34, 95% CI: 0.13–0.77) or overweight/obese (AOR: 0.61, 95% CI: 0.17–0.69) were less likely to think that an obese child would be healthy as an adult. Mothers of overweight/obese children were less likely to think that family members would be critical if their child was thin (AOR: 0.46, 95% CI: 0.23–0.87). 

Mothers with higher levels of education were more likely to think that some factors including lack of physical activity (AOR: 1.72, 95% CI: 1.00–2.95) and lack of outdoor playgrounds (AOR: 1.71 95%, CI: 1.02–2.90) would contribute to obesity. The perception about consuming junk food (AOR: 2.37, 95% CI: 1.43–3.96) and lack of playgrounds (AOR:1.61, 95% CI: 1.00–2.6) as a contributing factor for obesity was positively associated with high-income families. The differences between parents/mothers of boys and girls, and of fathers high and low education levels were small and not statistically significant in any of the adjusted models.

## 4. Discussion

This study aimed to understand maternal knowledge and perceptions of childhood obesity. We identified substantial knowledge gaps regarding perceptions of childhood obesity and its contributing factors among Bangladeshi mothers. We also noted significant underestimation of childhood obesity as well as specific socio-economic factors associated with maternal perceptions.

In Bangladesh, nearly all mothers stay at home and look after their children. Intervention programmes focusing on mothers could be effective if mothers are well educated about childhood obesity. In our study, nearly 65% of the mothers of preschool aged children were not aware of childhood obesity as a health problem. This lack of knowledge on childhood obesity presents a challenge to any intervention because an underlying cause of poor health (like obesity) might be perceived as a manifestation of good health. This is more relevant in developing countries like Bangladesh where undernutrition is prevalent and social norms dictate and shape such perceptions. This results in the co-existence of both under and overnutrition. A high level of misperceptions towards childhood obesity could exacerbate this situation.

In this study, we sought to understand whether maternal perceptions of childhood obesity as a health problem was related to other variables like family income, parental education, and the actual weight status of the children. We found that none of these factors were significantly associated with this perception. More than two thirds of the mothers were ignorant of any health consequences or risk factors of overweight/obesity irrespective of their education level. Mothers from developed (Italy, Japan) and middle-income countries (Malaysia, Iran) have reported being well-informed or knowledgeable about the consequences and risk factors of childhood overweight/obesity [19,20,21,22]. For instance, nearly 90% of the respondents in Malaysia and Iran were aware of the consequences of childhood obesity. Our study has revealed that only half of the mothers were aware of junk food as a risk factor of obesity. This lack of awareness can partly be due to the fact that undernutrition is a large child health problem in Bangladesh [4]. Most communication to mothers focuses on ensuring their children are eating enough food. As an example, preschool curricula or programmes include information about undernutrition whereas there is no information about obesity [23].

In our study setting, we found no statistically significant differences in terms maternal knowledge or perceptions of childhood obesity irrespective of preschool type, suggesting that perceptions or knowledge regarding childhood obesity are relatively homogenous in this country.

Maternal underestimation of their child’s weight status may result in unhealthy parenting practices [24]. We found a high level of maternal underestimation of childhood overweight/obesity. Interestingly, maternal recognition of normal and underweight children was accurate. Underestimation of overweight/obese status by their parents seems common internationally. For example, the level of underestimation is relatively higher among Asian (such as 24–40% in China, 98.98% in Taiwan, ~80%, in Malaysia, and 74% in India) and Latin American countries (such as 84% in Mexico and 65% in Brazil) [25]. However, most parents (87.5%) in South Korea estimated obesity status accurately [26]. Children who gain the most weight in adulthood are those whose weight statuses were underestimated during their childhood [27]. A lexicometric analysis concluded that education plays an important role in recognition of realistic weight statuses of children and promoting awareness regarding overweight/obesity [25].

We found around half of the participating mothers perceived that other family members would be critical if their child was slim. This perception seems entrenched in South Asian societies, as this part of the world is well known for its high rates of undernutrition [7]. In Bangladesh, a thin child is likely to be perceived by others as coming from a poor family. Mexican-American mothers were concerned about their thin children and perceived them as weak and at risk of becoming sick and even dying [28]. Similar perceptions might be prevalent in South Asian countries, like Bangladesh where child mortality rates have been high in the past decades [29].

More than half of the mothers perceived childhood overweight/obesity as a sign of good health. One reason for this misperception might be cultural prejudices. Mothers of skinny children are, in many cases, held guilty for not taking proper care of their children; thinness is often judged as proof of the mother’s negligence of responsibility. Because of a lack of knowledge, these mothers frequently consider their offspring’s thinness as analogous to parenting failure. As a result, parents are willing to see their children carrying excess weight. The perception of body image or weight is associated with many factors such as culture, geographic location, ethnicity, ethics and gender preference [11]. For example, Latina mothers tend to prefer a thin figure for themselves but a heavy figure for their children [30]. In Indian culture, overweight children are considered healthy if they are happy and loved [31].

In our study, there were no differences in perceptions of mothers whose child was a boy or girl. This finding is particularly important in the context of South Asian culture where favouring male children has been associated with undernutrition among female children in countries such as India [32]. It is likely that social perceptions and attitudes towards female gender are changing in Bangladesh due to various educational programmes. For example, the highly successful televised cartoon character, ‘Meena’ and other governmental and non-governmental developmental initiatives [33].

### Limitations

In this study, the sample preschools were selected using a stratified random sample procedure. Therefore, our sample is population representative. However, the target population was small since Jamalpur town is a small administrative unit compared to the entire population of preschools in Bangladesh. Bangladesh is a geographically small country with a large population which is homogeneous with respect to language, culture, and demographic characteristics as well as level of urbanisation. Therefore, it is likely that our findings could be generalisable to similarly sized district towns in other parts of Bangladesh. Weight status of mothers has been associated with the weight perception of their children. However, maternal anthropometric data was not included in this study.

## 5. Conclusions

From the perspective of a developing country, this study has revealed that the majority of the mothers did not consider childhood obesity as a health problem and most mothers had low levels of knowledge about the factors influencing childhood obesity. Our study has also suggested social factors which might be linked existing maternal perceptions of childhood obesity. Parents, particularly mothers, play a central role. Parent-centric educational interventions, focusing on mothers, should be a high priority for preventing childhood obesity in Bangladesh. Moreover, our study findings would contribute to adopt multisectoral approaches and to revise policies regarding health and nutrition literacy, unhealthy food and beverage tax, recommendations on the marketing of unhealthy foods to children, accessibility of unhealthy foods, food labelling, allocation of safe and accessible spaces for physical activity and advocating guidelines on healthy sleep and screen behaviours.

## Figures and Tables

**Table 1 ijerph-16-00202-t001:** Characteristics of participants.

Variables	Secular Preschool (*n* = 397) % (*n*)	Noorani Preschool (*n* = 188) % (*n*)	Total (*n* = 585) % (*n*)	*p*-Value
Age average (SD)	28.61 (4.57)	28.09 (4.76)	28.44 (4.64)	0.21 (a)
**Education**				0.03 (b)
Primary	16.9 (66)	20.7 (39)	18.13 (105)	
Secondary	27.1 (106)	36.2 (68)	30.05 (174)	
Higher secondary	25.6 (100)	21.3 (40	24.18 (140)	
Graduate	30.4 (119)	21.8 (41)	27.63 (160)	
**Occupation**				0.24 (f)
Housewife	85.5 (336)	90.4 (170)	87.09 (506)	
Service	13.2 (52)	8.5 (16)	11.7 (68)	
Business	1.3 (5)	1.1 (2)	1.2 (7)	
**Marital Status**				0.57 (cc)
Living with husband	98.0 (385)	96.8 (182)	97.59 (567)	
Single/separated	2.0 (8)	3.2 (6)	2.4 (14)	
**Husband’s education**				0.10 (b)
Primary	11.4 (44)	14.1 (26)	12.26 (70)	
Secondary	21.2 (82)	28.6 (53)	23.64 (135)	
Higher secondary	26.4 (102)	24.9 (46)	25.92 (148)	
Graduate	40.9 (158)	32.4 (60)	38.18 (218)	
**Husband’s occupation**				0.21 (f)
Business	37.8 (147)	35.1 (65)	36.93 (212)	
Day labour	3.9 (15)	7.0 (13)	4.88 (28)	
Farmer	2.6 (10)	2.7 (5)	2.61 (15)	
Other	1.5 (6)	3.8 (7)	2.26 (13)	
Service	54.2 (211)	51.4 (95)	53.31 (306)	
**Family income (monthly)**				0.04 (b)
BDT < 10,000	32.5 (127)	41.9 (78)	35.53 (205)	
BDT 10,000–24,999	26.1 (102)	27.4 (51)	26.52 (153)	
BDT 25,000–49999	31.5 (123)	21.0 (39)	28.08 (162)	
BDT > 50,000	10.0 (39)	9.7 (18)	9.88 (57)	
**Number of living children**				0.40 (cc)
1–2 kids	83.5 (329)	80.3 (151)	82.5 (480)	
More than 2 kids	16.5 (65)	19.7 (37)	17.5 (102)	
**Mode of birth delivery of child**				0.05 (cc)
Caesarean	58.4 (227)	49.5 (93)	55.5 (320)	
Vaginal	41.6 (162)	50.5 (95)	44.5 (257)	

a: *t*-test, b: Pearson’s Chi-square test without continuity correction, cc: Pearson’s Chi-square test with continuity correction, f: Fisher’s exact test for count data.

**Table 2 ijerph-16-00202-t002:** Actual and perceived prevalence of childhood overweight and obesity.

	Secular (*n* = 397) %, (*n*)	Noorani (*n* = 188) %, (*n*)	Total (*n* = 585) %, (*n*)	*p*-Value
**Actual weight status**				
Underweight	30.4 (120)	30.5 (57)	30.4 (177)	0.99
Normal	53.9 (213)	58.8 (110)	55.5 (323)	0.307
Overweight	5.8 (11)	5.9 (11)	5.8 (34)	0.99
Obese	9.9 (39)	4.8 (9)	8.2 (48)	0.056
Overweight and obese	15.7 (50)	11.7 (20)	14 (82)	0.971
**Perceived weight status**				
Underweight	31.6 (124)	37.9 (69)	33.6 (193)	0.165
Normal	65.3 (256)	58.8 (107)	63.2 (363)	0.157
Overweight and obese	3.1 (12)	3.3 (6)	3.1 (18)	0.99

**Table 3 ijerph-16-00202-t003:** Perceptions and knowledge of childhood overweight and obesity.

Variables	Yes % (*n*)	No % (*n*)	Dont Know % (*n*)
**Maternal perception**			
Childhood obesity is a health problem	35.23 (205)	10.57 (62)	54.04 (314)
Childhood obesity is a sign of good health	34.48 (200)	25.17 (146)	40.34 (234)
An obese child will be healthy when becomes adult	10.26 (60)	11.79 (69)	76.26 (452)
**Factors contributing childhood obesity**			
Consuming junk food	22.05 (228)	38.29 (224)	39.24 (228)
Lack of physical activity	38.55 (224)	18.93 (110)	42.51 (247)
Spending more than 2 h screen time	29.78 (173)	26.68 (155)	43.55 (253)
Lack of play ground	43.7 (254)	19.14 (11)	37.05 (175)
Knowledge of any health consequences of childhood obesity	31.42 (142)		68.58 (443)

**Table 4 ijerph-16-00202-t004:** Association of maternal perception (YES) in regards to various socio-economic factors and perceived weight status.

Characteristics	Childhood Obesity is a Health Problem (95% CI)		Childhood Obesity is a Sign of Good Health (95% CI)		An Obese Child will be Healthy When Becomes Adult (95% CI)		Family Member will be Critical if Child is Thin (95% CI)	
	OR	AOR	OR	AOR	OR	AOR	OR	AOR
Sex								
Boys	1	-	1	-	1	-	1	-
Girls	0.72 (0.5–1.02)	0.70 (0.46–1.07)	1.11 (0.76–1.62)	1.12 (0.71–1.76)	0.88 (0.51–1.52)	0.66 (0.32–1.28)	1.28 (0.92–1.78)	1.17 (0.77–1.75)
Mother’s education								
PS	1	-	1	-	1	-	1	-
HG	1.41 (1.00–2.00)	1.11 (0.65–1.92)	1.03 (0.71–1.50)	1.13 (0.63–2.06)	0.59 (0.34–1.00)	0.86 (0.35–2.09)	0.63 (0.46–0.88)	1.16 (0.69–1.98)
Father’s education								
PS	1	-	1	-	1	-	1	-
HG	1.23 (0.86–1.77)	1.02 (0.57–1.83)	0.96 (0.65–1.44)	0.97 (0.52–1.80)	0.42 (0.24–0.71)	0.62 (0.25–1.45)	0.53 (0.38–0.75)	0.66 (0.37–1.15)
Monthly income								
>10 K	1	-	1	-	1	-	1	-
25–50+ K	1.46 (0.98–2.20)	1.44(0.88–2.38)	0.7 (0.46–1.10)	0.69 (0.40–1.18)	0.35 (0.17–0.68)	0.42 (0.18–0.94)	0.49 (0.33–0.72)	0.64 (0.39–1.03)
Child’s status								
Underweight	0.97 (0.65–1.42)	1.27 (0.79–2.04)	0.91 (0.60–1.38)	0.83 (0.49–1.37)	0.44 (0.21–0.85)	0.34 (0.13–0.77)	1.7 (1.18–2.48)	1.36 (0.86–2.13)
Normal	1	-	1	-	1	-	1	-
Overweight/obese	1.16(0.70–1.92)	1.3 (0.70–2.36)	0.52 (0.26–0.96)	0.56 (0.25–1.14)	0.54 (0.20–1.23)	0.61 (0.17–0.69)	0.54 (0.31–0.91)	0.46 (0.23–0.87)

**Table 5 ijerph-16-00202-t005:** Association of maternal perception (YES) in regards to various socio-economic factors and perceived weight status.

Characteristics	Consuming Junk Food (95% CI)		Lack of Physical Activity (95% CI)		Spending More than 2 h Screen Time (95% CI)		Lack of Playground (95% CI)	
	OR	AOR	OR	AOR	OR	AOR	OR	AOR
Sex								
Boys	1	-	1	-	1	-	1	-
Girls	1.21 (0.86–1.70)	1.13 (0.73–1.74)	1.01 (0.72–1.42)	1.02 (0.67–1.55)	0.79 (0.55–1.14)	0.8 (0.52–1.24)	1.02 (0.73–1.43)	1.1 (0.73–1.65)
Mother’s education								
PS	1	-	1	-	1	-	1	-
HG	3.05 (2.15–4.36)	1.94 (1.11–3.38)	3.57 (2.51–5.13)	1.72 (1.00–2.95)	1.84 (1.28–2.66)	1.27 (0.72–2.23)	2.22 (1.59–3.11)	1.71 (1.02–2.90)
Father’s education								
PS	1	-	1	-	1	-	1	-
HG	2.64 (1.82–3.89)	1.08 (0.57–1.98)	3.06 (2.09–4.53)	1.45 (0.79–2.62)	1.67 (1.13–2.47)	1.17 (0.63–2.15)	2.28 (1.59–3.27)	1.3 (0.74–2.29)
Monthly income								
>10 K	1	-	1	-	1	-	1	-
25–50+ K	3.42 (2.27–5.23)	2.37 (1.43–3.96)	2.51 (1.68–3.79)	1.59 (0.97–2.60)	1.71 (1.13–2.63)	1.37 (0.82–2.31)	2.24 (1.52–3.33)	161 (1.0–2.60)
Child’s status								
Underweight	0.8 (0.54–1.18)	0.87 (0.52-1.42)	0.75 (0.51–1.11)	0.81 (0.50–1.31)	0.96 (0.63–1.46)	0.97 (0.58–1.59)	0.83 (0.57–1.21)	0.87 (0.55–1.39)
Normal	1	-	1	-	1	-	1	-
Overweight/obese	2.2 (1.35–3.63)	1.66 (0.89–3.08)	1.91 (1.17–3.14)	1.5 (0.82–2.76)	2.23 (1.35–3.68)	1.79 (0.97–3.25)	1.22 (0.75–1.98)	0.81 (0.44–1.47)

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
