# Peer review of "Is Childhood Overweight/Obesity Perceived as a Health Problem by Mothers of Preschool Aged Children in Bangladesh? A Community Level Cross-Sectional Study"

_ijerph, 2019, doi:10.3390/ijerph16020202_

Reviewer 1 Report

Dear Authors,

Let me thank you for allowing me to read your great work.

It is a very good study. After a careful reading, I found this paper is well balanced in literature review, methodology, statistics, discussion, and references. The limitation of the study are already identified and acknowledged in the paper. I have a very few minor suggestions to the authors which are written below. Other than that I would recommend this paper for the publication.

·         Line 41, despite significant progress, - can you add some information about progress in what with a reference.

·         Children at Secular schools have relatively higher rate of overweight/obesity than Madrasas. Readers in western countries would be interested more in knowing how these two setup differ. Some information is already provided but I would encourage authors to highlight their functioning little bit more.

·         Line 129, CDC’s LMS – I think it would be better for authors to mention full form of LMS (i.e. Lambda-mu-sigma / Learning management System?).

·         Tables – variables in all tables can be aligned to left. The difference between major and sub variables can indicated by larger font, italicization etc.

Reviewer

December 7, 2018

Author Response

Reviewer# 1

Let me thank you for allowing me to read your great work.It is a very good study. After a careful reading, I found this paper is well balanced in literature review, methodology, statistics, discussion, and references. The limitation of the study are already identified and acknowledged in the paper. I have a very few minor suggestions to the authors which are written below. Other than that I would recommend this paper for the publication.

Response: We would like to thank the reviewer for his/her thoughtful comments and suggestions for improving the manuscript quality. In the following section, we have addressed reviewer’s queries and recommendations. All changes made are highlighted using track change mode.

Line 41, despite significant progress, - can you add some information about progress in what with a reference.

Response: We have revised our manuscript as suggested by the reviewer (Page 2, Line: 41-42):

“In 1989-1990, the prevalence of underweight was 65% among children of under five years of age [4].”

Children at Secular schools have relatively higher rate of overweight/obesity than Madrasas. Readers in western countries would be interested more in knowing how these two setup differ. Some information is already provided but I would encourage authors to highlight their functioning little bit more.

Response:  We have revised our manuscript as suggested by the reviewer as follows (Page 2-3, Line: 78-85):

“The target population was all the children aged 4 to 7 years attending preschools in the municipality of Jamalpur. In Bangladesh, this includes playgroups, nurseries, kindergarten-I (KG-I) and KG-II.  The study population included secular preschools, which are usually a part of primary school and Noorani Madrasas- religious preschools equivalent to preschools. Unlike secular counterparts, most Noorani Madrasas are not a part of primary school. Families with relatively higher socio-economic backgrounds prefer secular preschools for their children, while highly conservative Muslim families prefer Noorani Madrasa preschools. Tuition fees in religious schools are nominal (often free of cost), whereas studying in the secular schools incur varying levels of tuition fees.”

 Line 129, CDC’s LMS – I think it would be better for authors to mention full form of LMS (i.e. Lambda-mu-sigma / Learning management System?).

Response:  Revised as recommended by the reviewer (Page 4, Line: 165-166).

Tables – variables in all tables can be aligned to left. The difference between major and sub variables can indicated by larger font, italicization etc.

Response: All Tables are revised as recommended by the reviewer (Page 5-7).   

Reviewer 2 Report

The data in this paper provides valuable information pertaining to maternal perception of childhood obesity in Bangladesh.  This paper is lacking a real connection to how this data can be used to move the needle.  More attention needs to be on this and in general the flow of the paper and grammar and punctuation could be improved, particularly toward the end of the paper.

1.      Needs hypothesis based on primary and secondary (if any) outcomes. 

2. Please define age of participants and the grades.  The abstract states preschool but the Methods say kindergarten.  Is this all before primary school starts or is kindergarten in primary school, are the schools free to parents or is some type of tuition involved, etc.  Describe socioeconomic status of families from secular schools vs. Noorani.

3. This reviewer suggests combining variables and procedures into well defined parts and describing in more detail each.  For example ... Height and weight: explain all variables about how it was measured, when it was measured, who measured it and their credentials, etc, 2) questionnaires: were any validated or have they been used before and in what population, etc, 3) xxx.  Include manufacturer information on all equipment.

4. Were any anthropometrics measured in mothers?  This should be added to the manuscript including appropriate statistical analysis to include data as appropriate ... another column or asterisk with appropriate detail, etc

5. Please be more specific about how you calculated sample size?  Did you use a tool?  Please give details.

6. Table 1 and 3. Relevant statistical tests need to be ran between Secular and Noorani schools and details about those stats need to be described.  Line 219-222 information needs to be more clear through this and stated earlier in the paper.

7. line 154, take out the word also

8. Discussion: restate aims and how this study addressed those.  

9. line 237, correct typo

10. line 238-239, correct punctuation

11.  line 243, suggest rewording appropriately.  Made to feel guilty, held accountable, etc

12. The discussion needs to be revisited.  There are more grammatical and punctuation errors than in the rest of the paper, and the flow could be improved.  Please include not collecting maternal anthropometric and/or socioeconomic data as a limitation if they were not assessed.  Please include how this data can be used to help with maternal perception of childhood obesity.  

Author Response

Reviewer#2

Comments and Suggestions for Authors

The data in this paper provides valuable information pertaining to maternal perception of childhood obesity in Bangladesh.  This paper is lacking a real connection to how this data can be used to move the needle.  More attention needs to be on this and in general the flow of the paper and grammar and punctuation could be improved, particularly toward the end of the paper.

Response: We would like to thank the reviewer for his/her thoughtful comments and suggestions for improving the manuscript quality. In the following section, we have addressed reviewer’s queries and recommendations. All changes made are highlighted using track change mode.

1.   Needs hypothesis based on primary and secondary (if any) outcomes.

Response:  We have revised our manuscript as suggested by the reviewer (Page 2, Line: 65-68):  

“We hypothesized that more than 20% of the mothers would consider childhood obesity as a health problem and this maternal perception would differ by school type. Our secondary hypothesis was that, there would be difference between perceived and actual overweight/obesity status of children by their mothers.”

Accordingly, result section has been revised (Page 7, Line: 203-205) as follows:

“Just over one third (35.04%) of respondents perceived that childhood overweight/obesity could be a health problem for their children, which was significantly higher than our hypothesis of 20% (Chi-squared = 84.82, p < 0.001). This perception did not differ between two school types (Chi-square = 0.38, p = 0.5) (Table S1).”  

2. Please define age of participants and the grades.  The abstract states preschool but the Methods say kindergarten.  Is this all before primary school starts or is kindergarten in primary school, are the schools free to parents or is some type of tuition involved, etc.  Describe socioeconomic status of families from secular schools vs. Noorani.

Response:  We have revised our manuscript as suggested by the reviewer (Page 2-3, Line: 78-85) as follows:

“The target population was all the children aged 4 to 7 years attending preschools in the municipality of Jamalpur. In Bangladesh, this includes playgroups, nurseries, kindergarten-I (KG-I) and KG-II.  The study population included secular preschools, which are usually a part of primary school and Noorani Madrasas- religious preschools equivalent to preschools. Unlike secular counterparts, most Noorani Madrasas are not a part of primary school. Families with relatively higher socio-economic backgrounds prefer secular preschools for their children, while highly conservative Muslim families prefer Noorani Madrasa preschools. Tuition fees in religious schools are nominal (often free of cost), whereas studying in the secular schools incur varying levels of tuition fees.”

3. This reviewer suggests combining variables and procedures into well defined parts and describing in more detail each.  For example ... Height and weight: explain all variables about how it was measured, when it was measured, who measured it and their credentials, etc, 2) questionnaires: were any validated or have they been used before and in what population, etc, 3) xxx.  Include manufacturer information on all equipment.

Response:  We have revised our manuscript in the Method section as recommended by the reviewer (Page 2-4, Line 104-157).

4. Were any anthropometrics measured in mothers?  This should be added to the manuscript including appropriate statistical analysis to include data as appropriate ... another column or asterisk with appropriate detail, etc

Response: We have not assessed anthropometric measurement for mothers.

5. Please be more specific about how you calculated sample size?  Did you use a tool?  Please give details.

Response: Based on reviewer’s suggestions, we have revised our manuscript (Page 3, Line 95-102) as follows:  

“To obtain an estimate of the proportion of mothers indicating that childhood obesity was a problem within 5% margin of error with 99% confidence, a sample size of 470 was required. The sample size was calculated using R package samplesizes4surveys [13]. For this, a two-stage complex design with 53 clusters (schools) in the population with approximately 1600 students, 20% of mothers being aware of childhood obesity as a problem and an intraclass correlation coefficient of 0.02 (i.e., design effect of 1.6) were assumed. To obtain the minimum sample size required, 8 secular schools and 4 Noorani Madrasas were selected according to probability proportional to the stratum size.”

6. Table 1 and 3. Relevant statistical tests need to be ran between Secular and Noorani schools and details about those stats need to be described.

Response:  p values are added in Table1 as suggested by the reviewer (Page 5-6).  Statistical tests between secular and Noorani schools were performed and described in the result (Page 2, Line 208-211). One Table S1 has been included as a supplementary table highlighting the difference between secular and Noorani preschools (Page 16).

Line 219-222 information needs to be more clear through this and stated earlier in the paper.

Response:  We have revised as recommended by the reviewer. This is as follows (Page 11, Line 266-270):

“In our study setting, we found no statistically significant differences in terms maternal knowledge or perceptions of childhood obesity irrespective of preschool type, suggesting that perceptions or knowledge regarding childhood obesity are relatively homogenous in this country.”

7. line 154, take out the word also

Response: Corrected  accordingly.

8. Discussion: restate aims and how this study addressed those. 

Response:  We have revised the manuscript accordingly to address the reviewer’s comment (Page 11, Line 235-236), which is as follows:  

“This study aimed to understand the maternal knowledge and perceptions of childhood obesity,   perception of weight status of their children and some socio-demographic factors associated with maternal perceptions.”  

9. line 237, correct typo

Response: Corrected  accordingly.

10. line 238-239, correct punctuation

Response: Corrected  accordingly.

11.  line 243, suggest rewording appropriately.  Made to feel guilty, held accountable, etc

Response: We have revised as recommended by the reviewer (Page 12, Line 293).

12. The discussion needs to be revisited.  There are more grammatical and punctuation errors than in the rest of the paper, and the flow could be improved. 

Response: We have revised the discussion section significantly as suggested by the reviewer,  (Page 11-12, Line: 234-330).

Please include not collecting maternal anthropometric and/or socioeconomic data as a limitation if they were not assessed. 

Response We have included this in the limitation section in the revised version of our manuscript (Page12, Line: 314-316) as follows:

“Weight status of mothers has been associated with the weight perception of their children. However, maternal anthropometric data was not included in this study.”   

Please include how this data can be used to help with maternal perception of childhood obesity. 

We have revised the conclusion part as to address the reviewer’s recommendation, which is follows (Page 12, Line: 322-330):

“From the perspective of a developing country, this study has revealed that the majority of the mothers did not consider childhood obesity as a health problem and most mothers had low levels of knowledge about the factors influencing childhood obesity. Our study has also suggested some social factors which might be associated with the poor perception of the mothers regarding childhood obesity. Parents, particularly mothers, play the most important central role in the prevention of childhood obesity. So parent-centric educational interventions, focusing on mothers, should be a high priority for preventing childhood obesity in Bangladesh. Moreover, our study findings would contribute to adopt multisectoral approaches and to revise policies regarding health and nutrition literacy, unhealthy food and beverage tax, recommendations on the marketing of unhealthy foods to children, accessibility of unhealthy foods, food labelling, allocation of safe and accessible spaces for physical activity and advocating guidelines on healthy sleep and screen and behaviours.”